## Original Research Article

classification; invasive species; *Lantana camara*; pollen viability; quantitative phase imaging.

V.K. and N.G. contributed equally to this work.

**Author for correspondence:**
Gitanjali Yadav,
E-mail: gy@nipgr.ac.in

# Quantification of pollen viability in *Lantana camara* by digital holographic microscopy

Vipin Kumar[1], Nishant Goyal[2], Abhishek Prasad[1], Suresh Babu[3], Kedar Khare[4] and Gitanjali Yadav[1]

[1]Biodiversity Informatics Laboratory, National Institute of Plant Genome Research, New Delhi, India; [2]Department of Physics, Indian Institute of Technology Delhi, New Delhi, India; [3]School of Human Ecology, Dr. B. R. Ambedkar University Delhi, New Delhi, India; [4]Optics and Photonics Centre, Indian Institute of Technology Delhi, New Delhi, India

## Abstract

Pollen grains represent the male gametes of seed plants and their viability is critical for sexual reproduction in the plant life cycle. Palynology and viability studies have traditionally been used to address a range of botanical, ecological and geological questions, but recent work has revealed the importance of pollen viability in invasion biology as well. Here, we report an efficient visual method for assessing the viability of pollen using digital holographic microscopy (DHM). Imaging data reveal that quantitative phase information provided by the technique can be correlated with viability as indicated by the outcome of the colorimetric test. We successfully test this method on pollen grains of *Lantana camara*, a well-known alien invasive plant in the tropical world. Our results show that pollen viability may be assessed accurately without the usual staining procedure and suggest potential applications of the DHM methodology to a number of emerging areas in plant science.

## 1. Introduction

Pollen grains commonly appear as fine dust; each grain represents a minute body of varying form and size, produced in specialised male floral organs called stamens in seed bearing plants, and transported to the female structures for fertilisation. Viability refers to the capacity of pollen grains to mature and then fertilise, followed by the ability to develop into seed and fruit (Faegri et al., 1989; Shapiro et al., 1965). Pollen analysis has long been known as a rigorous scientific method encompassing a diverse number of research disciplines including botany, paleontology, geology, ecology, climatology and archaeology (Dafni & Firmage, 2000; Faegri et al., 1989). More recently, in the modern era, pollen have been recognised as vectors for transgene escape from genetically modified crops (Firmage & Dafni, 2000). Furthermore, these tiny male reproductive units of seed plants have also been recognised for their importance in biological invasions (Bufford & Daehler, 2014). Ecosystem threats by invasive plant species often pose a huge challenge, not only for conservation but also for the nursery, horticultural and landscape industries, leading to the necessity of exploring sterile non-invasive cultivars to replace fertile invasive ones (Burns et al., 2019). In this work, we focus on the importance of novel imaging techniques in plant biology, to improve conventional pollen viability estimations (Alexander, 1969; Stanley & Linskens, 1974). We demonstrate an emerging quantitative phase imaging modality as a successful strategy for pollen viability estimations in *Lantana camara*, an aggressive plant invader, using our digital holographic microscopy (DHM) system that uses a novel single-shot sparse optimisation-based phase recovery (Khare et al., 2013; Mangal et al., 2019; Singh & Khare, 2017, 2018). We imaged 500 pollen grains in bright-field mode as well as in the quantitative phase mode by switching illumination without disturbing the samples, allowing us to establish phase map characteristics of viable and non-viable pollen in a quantitative manner, to improve upon existing pollen imaging literature in environmental sciences (Berg & Videen, 2011; Kemppinen et al., 2020; Sauvageat et al., 2020; Van Hout & Katz, 2004; Wu & Ozcan, 2018). We further provide initial evidence that quantitative phase mode of imaging may enable pollen imaging and classification without the need of conventional staining procedures.

## 2. Pollen viability in invasion biology

Pollen viability comprises different aspects of pollen performance such as fertilisation ability, germinability and stainability (Dafni & Firmage, 2000). A viable pollen is central to species dispersal, fitness and survival of the next plant generation. It is also essential for directed plant breeding and, consequently, crop improvement. Our interest in pollen analysis is an extension of our ongoing efforts to understand invasive alien plants species, particularly *Lantana camara* (Wild Sage) (Chauhan et al., 2022; Davis & Thompson, 2000; P. Mishra et al., 2021). Lantana is a small broad-leaf flowering shrub within the Verbenaceae family, native to Central and South American tropics, but now dominating landscapes in over 50 countries, making it one of the world's top 10 invasives (Ghisalberti, 2000; Sharma et al., 1988). Pollen viability data become important in the context of both, species invasivity and habitat invasibility (Jiang et al., 2022). Other studies have also established that assessment of morphological and cytological differences among *Lantana* varieties can help in measuring invasive potential and suitability for commercial production and landscape use (Steppe et al., 2019).

Pollen viability can be affected by drought/dehydration, heat stress and UV-B radiation. These factors can play a role after pollen dehiscence, when pollen is exposed to the environment, or even before, during pollen development inside the anther. Viability may also be species-specific and dependent on pollen physiology, or the presence of specific structural modifications. A complicating factor in these assessments is the lack of standardised protocols and experimental conditions for viability assessments as described below (Dafni & Firmage, 2000; Firmage & Dafni, 2000).

### 2.1. Drawbacks of conventional viability estimations

As pollination is the primary function of the pollen grain in a plant life cycle, one way to test pollen viability is to use the pollen for pollination and subsequently analyse the seed set. However, this is time-consuming and often not feasible; thus, other methods are frequently used to elucidate pollen viability, such as staining techniques, in vitro or in-vivo germination, as well as semi-in situ germination on the excised stigma (stigmatic germination), or impedance flow (IF) cytometry, but again, most of these are difficult to scale up and further confounded by incompatibilities, post-fertilisation barriers and limited measurability, factors that restrict the accuracy of these tests (Dafni & Firmage, 2000; Dionne & Spicer, 1958).

Vital staining is by far the fastest and most commonly used method of pollen viability estimation, but the last major advance in this area was almost half a century back when the Heslop-Harrisons developed a viability test based on fluoro-chromatic reaction-based membrane integrity and enzyme activity (Heslop-Harrison, 1977; J. Heslop-Harrison & Heslop-Harrison, 1970; Shivanna & Heslop-Harrison, 1981). In general, pollen staining involves visualisation of specific compounds, contents or cellular compartments that take up the dye colour, while the absence of colour indicates aborted pollen (Stanley & Linskens, 1974; Alexander, 1969). Confounding factors in this procedure include failures to discriminate different viability levels, the toxicity of dyes being a concern for human health apart from the dye stain rendering pollen grains unfit for germination (Ge et al., 2011). Refined viability estimations have been attempted by comparing and combining techniques like cytometry and staining in case of mature cucumber, sweet pepper and tomato pollen (Heidmann et al., 2016). However, pollen viability assessment purely through microscopy, as envisaged in this work, has not been investigated up till now.

### 2.2. Imaging of pollen grains and limitations of current methods

Traditionally, palynologists have used compound light microscopy for pollen identification and interpretation and scanning electron microscopy (SEM) for morphological comparisons and taxonomy (Jones & Bryant, Jr., 2007). Doubtless, SEM offers far greater resolution, but sample preparation and the time needed to count, analyse, photograph and print the micrographs, and the consequent lack of scalability are the limiting factors. Flow cytometry methods were then described to determine pollen viability based on the nuclear DNA content in mature pollen grains, but this is a fairly lengthy and expensive process not suitable for a quick estimate (Bino et al., 1990).

With the advancement of digital microscopy, palynology studies are becoming less time-consuming and can generate more reliable data for species taxonomy, apart from saving hours spent on manual counts of pollen grains following the process of staining to differentiate between viable and non-viable pollen (S. Mishra & Srivastava, 2015). At the turn of the millennium, alternative digital methods for counting pollen were devised, but often without regard to viable and non-viable grains, and these protocols also had drawbacks of software specific to branded instruments (Aronne et al., 2001; Bechar et al., 1997). More recently, particle counters and binary 2D projection, including shape-based Fourier descriptors and topological features, have been used, but the success varies by species, and is applicable only when size difference settings for viable and non-viable grains is larger than the natural variation in viable pollen size for that species (Akcam & Lohweg, 2022; Kelly et al., 2002; Mudd & Arathi, 2012).

In summary, digital image processing appears to overcome size-based differences and species limitations but has not been explored sufficiently widely, emphasising the importance of another simpler method for pollen gain imaging in natural/hydrated state for assessing viability, as described in this work by means of the DHM imaging methodology.

### 2.3. Digital holographic microscopy for pollen imaging

DHM is an interferometric imaging modality that uses a low-power (few milliwatts) coherent laser illumination in a Mach–Zehnder configuration. As shown in Figure 1a,b, the laser beam is first split in two. One of the beams is transmitted through the pollen sample and forms the image field $O(x,y)$ at the detector plane, while a second beam takes an identical path without any sample and forms a tilted plane reference $R(x,y)$. Since the two optical fields $O(x,y)$ and $R(x,y)$ have originated from the same coherent laser beam and travel near identical path lengths up to the detector, they are temporally coherent and produce interference fringes with good contrast at the detector. A low-power light source is sufficient for this modality as we are not exciting any fluorescence labels but simply transmitting the light beam through a semi or fully transparent sample. The interference pattern (or digital hologram) $H(x,y)$ recorded on the pixelated array detector may be mathematically described by

$$H(x,y) = |R(x,y) + O(x,y)|^2$$
$$= |R|^2 + |O|^2 + R^*O + RO^*. \quad (1)$$

The $*$ operation above is a complex conjugation of wave functions associated with the object and reference beams. Our DHM system implements image plane holography, where $O(x,y)$ represents the image field of the pollen recorded using a 40× infinity-corrected

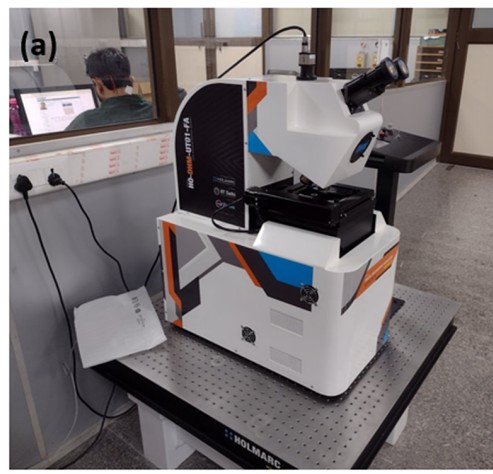
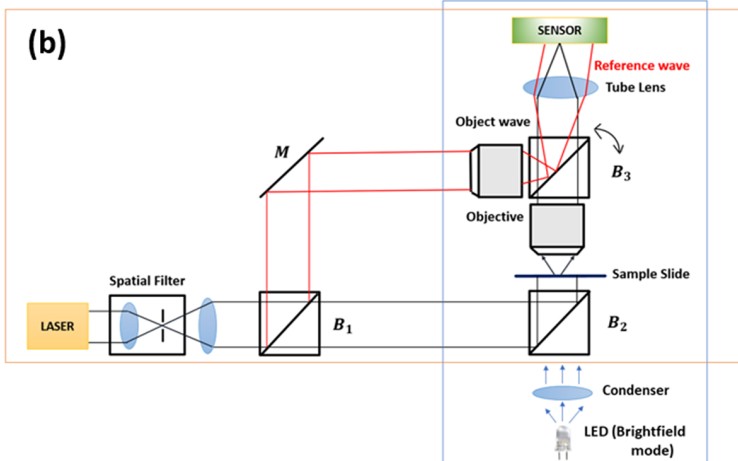

**Figure 1.** Digital holographic microscope system used in the present study. (a) System photograph. (b) Nominal optical layout of the system. $B_1, B_2, B_3$: beam-splitters; $M$: plane mirror. The beam-splitter $B_3$ can be rotated with an adjustable screw in order to introduce tilt in the reference wave as required for off-axis hologram recording. The system is fitted with a laser source for phase imaging and LED illumination with condenser for bright-field illumination. By switching between the two illuminations, one can record an image plane digital hologram or a bright-field image of the sample in the same position.

imaging system. The image field $O(x,y)$ at the detector plane is a complex wave function and contains both amplitude and phase parts. The amplitude part signifies the information regarding the attenuation or absorption of the illuminating light when passed through the pollen sample. The important feature of a DHM is that it is able to record and numerically reconstruct the phase $\phi(x,y)$ of the image field as well. The phase map $\phi(x,y)$ may be interpreted physically as a projection:

$$\phi(x,y) = \frac{2\pi}{\lambda} \int n(x,y,z)\,dz. \tag{2}$$

Here, $n(x,y,z)$ represents the relative refractive index in the 3D space of the pollen sample relative to the surrounding medium. In other words, the phase function $\phi(x,y)$ represents the optical path length for the illuminating laser beam through the sample pollen. Note that the refractive index $n(x,y,z)$ is an inherent material property of the pollen grain being imaged. This material property is thus recoverable from a DHM system but not accessible with a bright-field system. Note that the refractive index $n(x,y,z)$ has been integrated over the depth of the sample (or the propagation direction $z$ of the laser beam through the sample). For completeness, we remark that the more commonly used phase contrast mode in microscopes does utilise the phase property of light to enhance the qualitative visual appearance of the samples. We have provided comparative images of two pollen grains in bright-field, phase contrast and quantitative phase modes in Supplementary Figure S1. This supplementary illustration suggests that unlike DHM imaging, the traditional phase contrast information is qualitative in nature and does not bring out distinct morphological features of viable versus non-viable pollen.

We highlight here that DHM is a computational microscope. The phase information of interest is not directly measured on the array detector. Phase can, however, be estimated computationally using numerical processing of the recorded hologram pattern $H(x,y)$ at the detector plane. The speciality of our DHM system is that it provides single-shot phase imaging at full diffraction limited resolution. With current advances in the CMOS array sensor technology, a good contrast fringe pattern may be recorded in just a millisecond exposure time when used with a few-milliwatt laser source for illumination as in our case. The single-shot operation is

advantageous as the system is not very sensitive to the surrounding vibrations during the exposure time.

Single-shot off-axis holograms as in our study are conventionally reconstructed using the Fourier-transform method (Kreis, 1986; Takeda et al., 1982). This method inherently involves a low-pass filtering operation and, as a result, the recovered object wave function $O(x,y)$ often has lower resolution compared with the diffraction-limit of the infinity corrected imaging system used. In image plane holography, this loss of information results in blurring of edges in the image field that are critical to perception of image quality. Contrary to this, the bright-field images recorded using the same infinity corrected imaging system allow full diffraction-limited resolution. Hence, the phase information recovered using the Fourier-transform method has lower spatial resolution compared with the conventional bright-field images. Other methods such as phase-shifting methods (Creath, 1988; Yamaguchi, 2006) can deliver the expected full resolution, but require multiple hologram recording with much stringent vibration isolation, making the DHM system costs high. In order to retrieve the phase information with full pixel resolution from a single-shot off-axis hologram, we use a sparse optimisation method that we developed over last several years of effort (Khare et al., 2013; Mangal et al., 2019; Singh & Khare, 2017, 2018). The resolution and phase accuracy advantages (Singh et al., 2015) of this optimisation method are already well established in prior literature.

In brief, in the sparse optimisation methodology, the reconstruction of the complex object wave $O(x,y)$ is formulated as an optimisation problem where we minimise a cost function given as

$$\begin{aligned} C(O,O^*) &= C_1 + C_2 \\ &= \|H - (|R|^2 + |O|^2 + R^*O + RO^*)\|_2^2 + \alpha\,\psi(O,O^*). \end{aligned} \tag{3}$$

Here, the $\|...\|_2^2$ represents the squared L2-norm of the quantity inside. The first term in the cost function represents the squared data fitting error between the measured hologram data $H(x,y)$ and the interference pattern formation model (equation (1)), whereas the second term $\psi(O,O^*)$ represents a suitable image domain constraint. The constant $\alpha$ denotes a positive regularisation constant that determines the relative weight between the two terms of the cost function. In the present work, we used the total variation (TV)

penalty function as a constraint for the optimisation problem. TV is a popular edge-preserving penalty function that has been widely used in the imaging literature, which is suitable for the present image plane holography data. The TV penalty is defined as

$$\psi(O, O^*) = \iint |\nabla O(x,y)| \, dx \, dy, \tag{4}$$

where $\nabla$ represents the two-dimensional numerical gradient operation. Further, we utilise an adaptive alternating minimisation scheme inspired by Sidky and Pan (2008) for the reconstruction of the complex-valued object field $O(x,y)$ that does not involve empirical selection of any specific regularisation parameter $\alpha$. The adaptive alternating minimisation scheme is already described in prior publications (Mangal et al., 2019; Singh & Khare, 2017). The minimisation of cost $C(O, O^*)$ is achieved iteratively. Since the unknown object function $O(x,y)$ is complex-valued, Wirtinger derivatives of $C_1$ and $C_2$ with respect to $O^*$ are evaluated to obtain the corresponding steepest descent directions. The Wirtinger derivatives of two terms $C_1$ and $C_2$ are given as

$$\nabla_{O^*} C_1 = -2 \left[ H - |R + O|^2 \right] \cdot (R + O) \tag{5}$$

and

$$\nabla_{O^*} C_2 = -\nabla \cdot \left[ \frac{\nabla O}{\sqrt{|\nabla O|^2 + \epsilon^2}} \right]. \tag{6}$$

Here, $\epsilon^2$ is a small positive number ($\approx 10^{-10}$) used to prevent zero division. The essence of the optimisation process is that in each iteration, the numerical values of $C_1$ and $C_2$ are reduced in an adaptive manner, such that the change in the solution due to the reduction in $C_1$ and $C_2$ is balanced. The iterative process is stopped when the solution $O(x,y)$ changes negligibly in successive iterations. Since this optimisation process operates completely in the image domain, a region of interest can be suitably chosen to allow a full-resolution recovery of $O(x,y)$ over a region of interest of the recorded image plane hologram $H(x,y)$. In other words, a user can easily select a region of interest near a pollen grain for the phase map recovery. In the present study, a $512 \times 512$ pixel area is used for phase reconstruction of individual pollen. Using the optimisation procedure, the complex-valued image field $O(x,y)$ is recovered with diffraction-limited resolution. The phase map $\phi(x,y)$ associated with this image field is defined as the ratio of

imaginary and real parts of $O(x,y)$:

$$\phi(x,y) = \arctan\left( \frac{Im \left[ O(x,y) \right]}{Re \left[ O(x,y) \right]} \right). \tag{7}$$

Since the arc-tangent function is defined only for the range $[-\pi, \pi]$, the phase map $\phi(x,y)$ retrieved from the optimisation procedure is in a wrapped form. In order to unwrap this phase map and derive the full physical information, a 2D unwrapping algorithm has been used that is based on the transport of intensity equation method (Pandey et al., 2016). The unwrapped phase function can be associated with the projection definition in equation (2).

## 3. Methodology

### 3.1. Sample collection and pollen analysis

**3.1.1. Sample collection and staining.** *Lantana camara* flowers (see Figure 2a) were collected from Hauz Khaz Rose Garden (28°32′52.09″N/ 77°11′27.77″E), which is a part of Delhi Ridge forest. Fresh flowers were plucked early morning, which were then directly transferred to the laboratory for analysis. Pollen grains were extracted from florets and transferred on slide using the tapping and squashing method. For stain preparation, 2 g of Carmine powder was dissolved in 95 ml of glacial acetic followed by an addition of distilled water to make a 100-ml solution. The solution was boiled, filtered and stored at room temperature. Flower clusters (inflorescences) were collected when the clusters each had at least one flower partially open. Pre-dehiscent anthers were removed from unopened flowers and placed in a 1.5-ml Eppendorf tube with $\approx$ 100 $\mu$L of acetocarmine stain. Anthers were stained and rinsed three times with distilled water. Care was taken not to burst the anthers while rinsing them. Rinsed anthers were squashed in 50 $\mu$L of glycerol (diluted with 20% water) and transferred onto a microscope slide and covered with a cover slip. The slide was gently tapped and pressed to release pollen grains out of anthers. Pollen viability was then observed immediately through microscopic imaging.

**3.1.2. Conventional viability estimation.** The grains were observed under a bright field Nikon Eclipse E200 upright microscope, using a 40× magnification objective for initial preparation. Photographs of pollen grains were taken using an IDS $\mu$Eye 3070 CP camera mounted on the microscope. Pollen grains stained to dark red were considered viable. As shown in Figure 2b, uniformly round, non-wrinkled, red coloured pollen grains were considered viable,

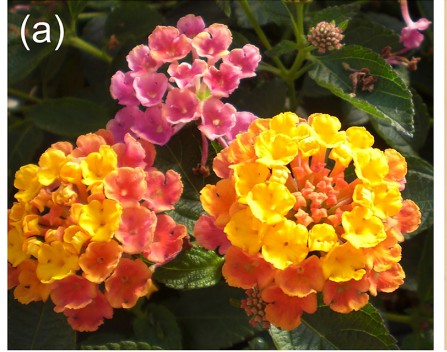
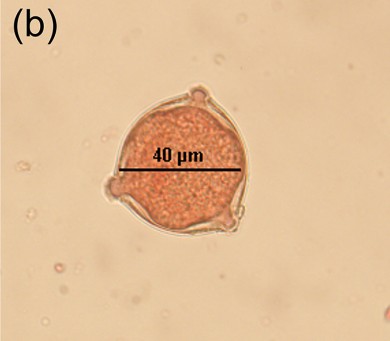
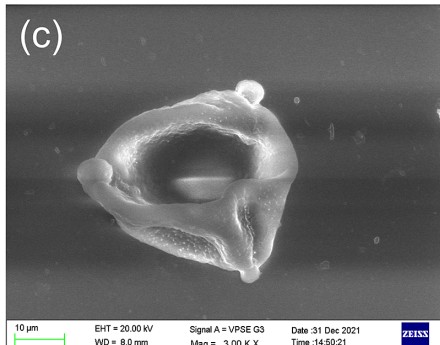

**Figure 2.** (a) *Lantana camara* flowers, and typical *Lantana* pollen grain images observed with (b) an optical microscope and (c) a 20-kV scanning electron microscope. Note the surface texture and tricolpate nature of the pollen grain.

whereas non-stained and wrinkled or deformed pollen grains were considered non-viable. Manual scanning of over 1,200 pollen grains in this manner and removal of clumped grains led to the selection of 500 clearly defined viable and non-viable pollen grains in nearly equal proportion.

### 3.2. Species validation via scanning electron microscopy

The surface texture of pollen grains was examined using a SEM (EVOLS10, Carl Zeiss) and this was used to validate species identity. Specimens of selected viable *Lantana camara* pollen grains were prepared for SEM by fixing the pollen in 2.5% (v/v) glutaraldehyde in a 0.1-M sodium phosphate buffer, vacuuming them thrice, then leaving them to dry for 24 hr. Samples were then coated with gold palladium using Sputter Coater SC7620, before observing under SEM. Images were captured using SmartSEM software and the images of the mounted pollen specimens were taken at bar 10 $\mu$m. All procedures were performed as per the protocol of the manufacturer. The pollen unit was found to be monads or tetrads, with psilate-type sculpture and colpi largely tricolpate, as shown in Figure 2c. Quantitative parameters such as polar diameter, mesocolpium distance, equatorial dimensions, aperture size, spine diameter and exine thickness can be calculated from these images using the open-source ImageJ software (https://imagej.nih.gov/ij/).

### 3.3. DHM imaging of Lantana pollen

The DHM system used in our study (Make: Holmarc Opto-Mechatronics, Kochi, India, Model: HO-DHM-UT01-FA) works in dual mode, that is, both bright-field and holographic, as depicted in Figure 1. It employs a Nikon Fluor 40X, 0.75NA objective lens for imaging. The system uses an array sensor (Make: IDS GmbH, model: $\mu$Eye CP-3070) with 3.4-$\mu$m square pixels. The user can switch between the laser and LED illuminations for recording an image plane hologram or a bright-field imaging of the sample without physically disturbing it. Since the phase imaging is not a familiar modality among plant science researchers, the bright-field mode can help locate the individual pollen over the sample slide, which may be focused appropriately. The holographic mode which essentially involves switching of the illumination can then record an image plane hologram of a pollen in the same focused position. Numerical reconstruction as described in Section 2.3 applied to a recorded hologram then provides the user with the quantitative phase information $\phi(x, y)$ associated with each pollen grain. Since the phase map represents optical path length, it is a common practice to render it as a surface plot to provide a sense of depth information regarding the object being imaged. The speciality of our DHM system is that it provides single-shot phase imaging at the full diffraction limited resolution.

## 4. Results

### 4.1. Quantitative phase imaging of Lantana pollen

For illustration, in Figure 3, we show the bright-field image, the recorded hologram (or interference pattern) and the 3D rendered phase map for a few select pollen of different types from 500 individual *Lantana pollen* images recorded by the DHM. The first column is the bright-field images that provide qualitative information on viable and non-viable pollen based on the dye stain. The second column represents the off-axis image plane hologram of the same pollen grain, with clearly visible bending of interference fringes at

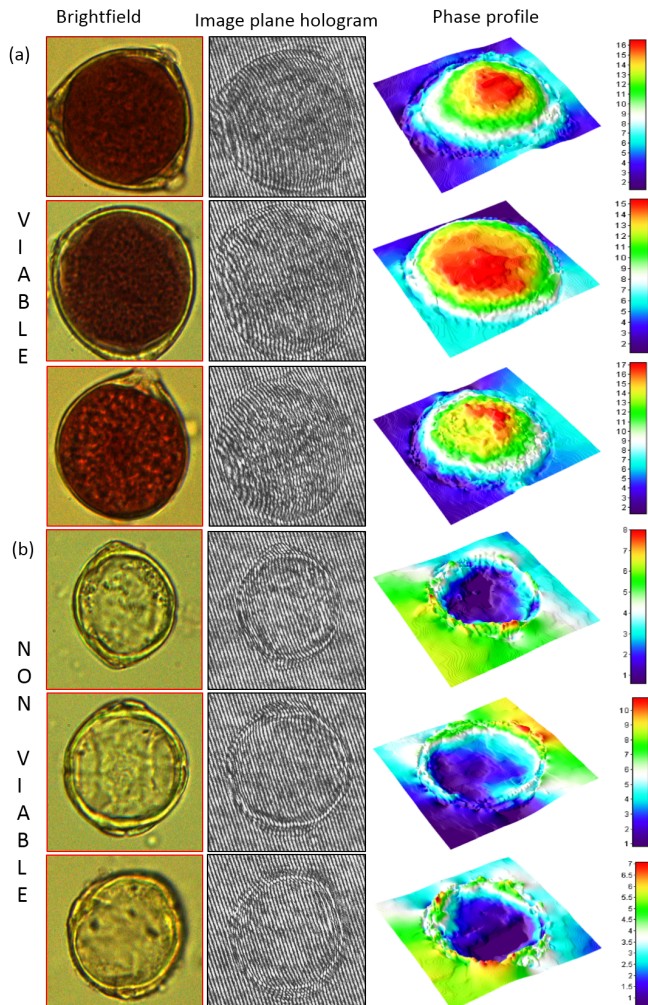

**Figure 3.** Bright-field image, recorded hologram and 3D rendered phase map for (a) viable pollen and (b) non-viable pollen. Note the ridges and the fissures in the image plain hologram that are reflected as distinct layers in the phase profile.

the location of the pollen grain. The corresponding DHM phase map derived from the fringe pattern is depicted in the rightmost column as a surface plot as the phase map represents the optical height map of the pollen grain. It is interesting to observe that for viable pollen grains (red in Figure 3a), the corresponding phase map has a nearly hemispherical shape. In contrast, the colourless (aborted) pollen shown in Figure 3b has a nearly flat phase map (also evident in the nearly straight line nature of the interference fringes within the pollen area). This observation agrees with the prior knowledge that vital stain binds to sugars and DNA that absorb the dye in the viable pollen and suggests that the quantitative phase maps provide a novel parameter for distinguishing viable and non-viable pollen grains. In Figure 3b, for a non-viable pollen, the quantitative phase values within the pollen appear to be lower than the surroundings, suggesting a negative refractive index contrast. This aspect needs further investigation in future. We note that the fringe bending due to phase shifts is of primary interest here and the phase map is not affected by any amplitude modulation of the interference fringes due to absorption of laser by the dye. It is known that the acetocarmine dye does not have significant absorption (Lima-De-Faria & Bose, 1954) at the laser wavelength 650 nm used in our experiment. The DHM-based classification of pollen grains can be further quantified by numerical measurements

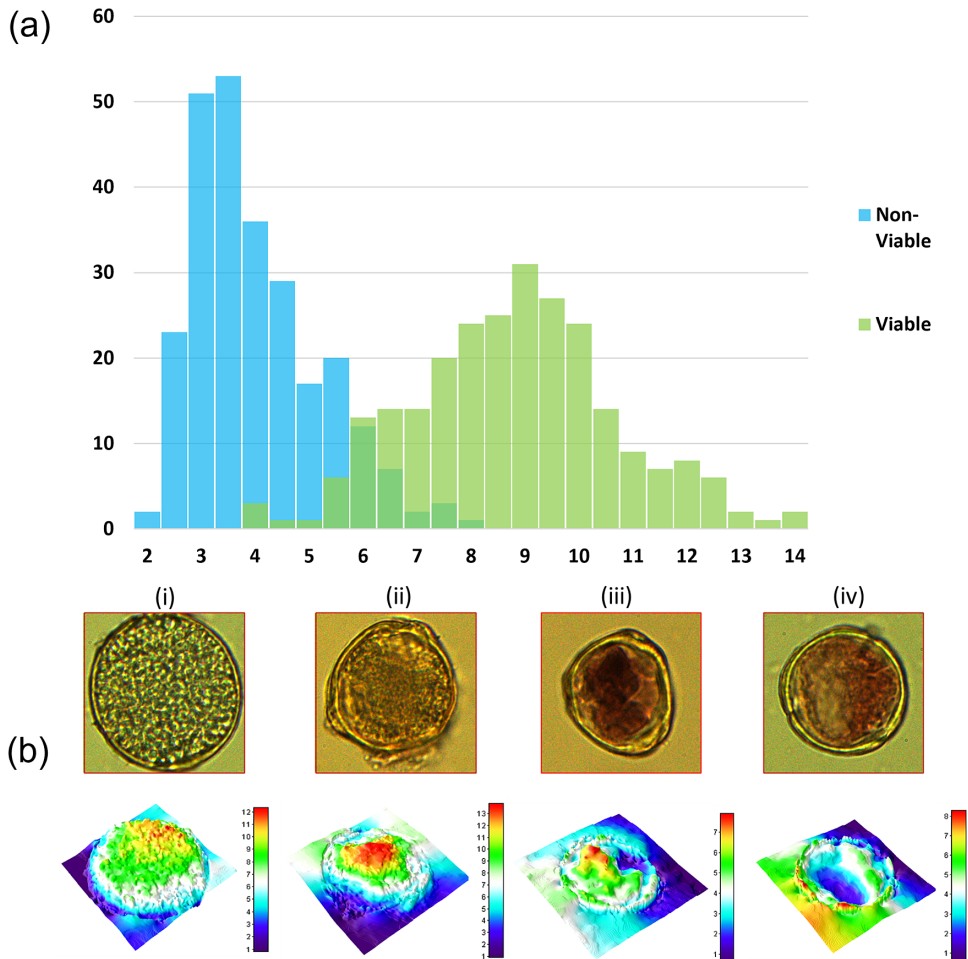

**Figure 4.** (a) Histogram of mean phase values for two different classes of pollen. The blue and green colours represent histograms for non-viable and viable pollen, respectively. (b) Typical pollen observed in the overlap region of the two histograms. Figures (i) and (ii) are cases where the mean phase value is high but the pollen do not show red colouration. Figures (iii) and (iv) are cases where red colouration is seen but the pollen phase map shows partially empty areas leading to a low mean phase value. The cases (i)–(iv) represent typical instances where viable/non-viable labelling based on colouration alone appears to be incorrect.

on pollen phase maps, and observing the corresponding statistics, as we demonstrate in the next section.

## 4.2. Quantitative measurements on bright-field and phase images

The dual-mode nature of our DHM system allows us to record the image of a single pollen in both bright-field and quantitative phase modes without disturbing the sample. Since the two images are recorded with the same array sensor, it is very easy to identify the same pollen in images from the two modalities as is evident in Figure 3. The images can be read in a standard open-source image processing platform like ImageJ. In this study, we collected 508 pollen images in both bright-field and quantitative phase modes. The bright-field image provides two-dimensional parameters associated with the pollen such as area and perimetre. The phase image, on the other hand, provides additional quantitative parameters such as mean phase and optical volume. In Figure 4a, we plot a histogram of the mean phase values of all pollen, obtained by dividing the integrated phase values within the pollen by the area of each pollen grain. The histogram is labelled with two colours corresponding to the viable and non-viable pollen (based on the presence or absence of dye stain in the bright-field images). The two distinct peaks in the histogram reveal that DHM is able to assess

**Table 1.** Statistics of mean phase values within the pollen area for non-viable and viable pollen

| Pollen type | No. of samples | Mean phase | Std. dev. |
|---|---|---|---|
| Non-viable | 256 | 3.90 | 1.24 |
| Viable | 252 | 9.01 | 2.17 |

pollen viability and pollen grains may be classified into accurate classes based on the phase map data. The statistics of the viable and non-viable pollen is described in Table 1. The two-sample $t$-test performed on the two classes of pollen has a $p$-value that is much lower than 0.05 suggesting clearly different class means. With the observation of clear distinction in phase maps for viable and non-viable *Lantana pollen* as described above, we further decided to examine the phase maps for unstained pollen. A fresh pollen sample was thus prepared without staining and the pollen were imaged with water as surrounding medium. Two illustrative pollen images for this unstained case are shown in Figure 5a,b. Traditionally, there is no straightforward method to distinguish these pollen in the bright-field mode in the absence of the red stain. Their corresponding phase maps, however, suggest a possibility of distinguishing them. The mean phase value for pollen in Figure 5a is 8.72 and that for pollen in Figure 5b is 4.26. The numerical

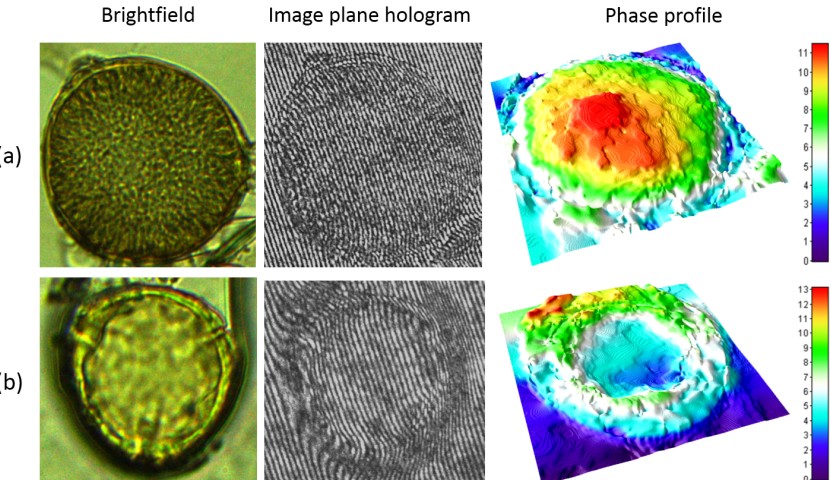

**Figure 5.** Illustrative bright-field image, image plane hologram and 3D rendered phase map for two unstained pollen samples. The top and bottom rows represent viable and non-viable pollen with mean phase values 8.72 and 4.26, respectively.

mean phase values are well within the statistics and histograms presented in Table 1 and Figure 4, respectively, for the two pollen classes. It may be noted that even after the imaging is performed, the viable pollen grains are not damaged and are available for further analysis. Additional data and detailed study will be required in future to establish a label-free pollen classification method based on quantitative phase imaging.

There is an overlap in the two distributions on the histogram, suggesting similar mean phase values for approximately 10% of total pollen grains (out of 508) that were classified visually as viable or non-viable nominally based on their red stain. Bright-field and phase images of four typical pollen in this overlap region are shown in Figure 4b. Figure 4b (i and ii) was labelled as non-viable based on lack of stain colouration, but shows significant phase values, suggesting that these pollen may be viable but somehow did not absorb the dye during our sample preparation. In contrast, Figure 4b (iii and iv) was labelled as viable due to their red colouration, but their phase maps show partially empty areas, leading to low mean phase values. These observation suggests that the colour-based classification in itself may have some problems that are revealed by DHM imaging. The correct classification of pollen in the overlap region of histogram needs more investigations.

Finally, we remark that a simple visual inspection-based method for pollen classification also seems to be possible without actual phase reconstruction. In particular, we note from Figures 3 and 5 that viable pollen contain additional material in the pollen volume and, as a result, the interference fringes appear curved within the pollen boundary. On the other hand, for non-viable pollen that have lost the cytoplasm material, the interference fringes are nearly straight lines within the pollen area. A method based on the observation of nature of fringes can become an interesting future topic to be explored. In the present study, we mainly used the mean phase values within the pollen area for distinguishing between the two pollen types. The phase images, however, clearly have much more information in the form of textural details that may be useful for further DHM imaging studies on pollen.

## 5. Conclusions and future directions

Pollen viability refers to the capacity of pollen to get mature and then fertilise, followed by development into seed and fruit. Quantification of viability enables estimation of male reproductive potential of entire species, cultivars and populations. However, as discussed in this paper, viability estimation can be a time-consuming, error prone and low scale process, even the fastest methods available today require overnight dye-based staining.

In this work, we describe a quantitative imaging technique using a single-shot full-resolution DHM system which is able to image pollen in a hydrated state. Not only do we demonstrate the successful classification of viable and non-viable pollen by the DHM, but our preliminary data reveal the possibility of quantifying viability of pollen from unstained pollen samples, thus requiring minimal wet-lab processing for sample preparation. The DHM system provided full diffraction-limited images of the same pollen grain in the bright-field as well as quantitative phase modes, allowing us to establish a correspondence between pollen that show (or do not show) colouration due to vital staining and their corresponding quantitative phase maps. The simplicity of our imaging methodology allowed us to image over 500 pollen grains quickly for making our findings statistically significant. We demonstrate the application of pollen viability to invasion biology by testing our method on pollen grains of the invasive *Lantana camara*, a well-known plant invader known to most of the tropical world. Low viability among invasive cultivars as indicated by the DHM can be used to identify the most suitable horticultural varieties for ornamental use and distribution.

There are several different directions in which this preliminary study can be continued further in invasion biology. At present, we have only used the mean phase from among 30 different morphological parameters that can be extracted from the DHM phase images. We plan to use other parameters for estimating C value content within the grain, through machine learning if possible, which in turn would provide a handle to determine/quantify ploidy levels. Such investigations can transform plant biology, especially for invasive species where variation in ploidy levels has been shown to be an important factor in invasivity. Polyploidy has created many possibilities in plant breeding, such as the production of improved cultivars, because of increased vigor, larger organs, higher yield levels, increased tolerance to biotic and abiotic stresses and production of seedless varieties (Sattler et al., 2016). There is presently no way to determine ploidy without extensive cytogenetics or flow cytometry, but it has been reported that ploidy levels in *Lantana* are primary determinants of pollen stainability/male sterility, with diploids exhibiting the highest pollen stainability, followed by

tetraploids, pentaploids and hexaploids (Czarnecki et al., 2014). This observation, when combined with the capacity of DHM to quantify cell components including DNA that pick up the vital stain, reveals *Lantana* to be a particularly suitable model species to explore ploidy-level determination via DHM. This, in turn, will require detailed investigation of cell components that are actually represented by phase map data, and is thus a future direction for our work.

In summary, the present work on establishing the effectiveness of quantitative phase imaging modality for pollen imaging thus opens up multiple new avenues for further research as we hope to report in future.

**Financial support.** We acknowledge support from the Central Research Facility, IIT Delhi for making the Digital Holographic Microscope system available for this work. We also acknowledge the support of the National Institute of Plant Genome Research (NIPGR), New Delhi, for infrastructure, the Confocal facility of NIPGR for the SEM work and DBT-eLibrary Consortium (DeLCON) for providing access to e-resources. G.Y. acknowledges the support of NIPGR, New Delhi, and funds from BBSRC GCRF Grant ID BBSRC BB/P027970/1TIGR2ESS for this work. V.K. was funded by the same BBSRC grant as a Post-Graduate Research Fellow (PGRF). N.G. acknowledges support from the Prime Minister's Research Fellowship (PMRF) for pursuing PhD thesis work at IIT Delhi. A.P. received the JRF and SRF fellowship from the Council of Scientific & Industrial Research (CSIR), Government of India as part of his PhD at NIPGR. K.K. acknowledges support from the Abdul Kalam Technology Innovation National Fellowship provided by the Indian National Academy of Engineering (INAE). The publication charge of this article is covered by NIPGR Core Grant. These funding bodies provided the equipment required for this work but do not have any role in design of the study and collection, analysis and interpretation of data and in writing the manuscript.

**Competing interest.** The authors have no competing interest.

**Author contribution.** Conceptualisation: K.K., G.Y.; Data collection: V.K., N.G., A.P.; Funding: G.Y., K.K., A.P.; Investigation: V.K., N.G., K.K., G.Y.; Supervision: K.K., S.B., G.Y. Visualisation: K.K., V.G., N.G., G.Y.; Writing—original draft: K.K., G.Y.; Writing—review and editing: K.K., A.P., S.B., G.Y. All authors have read and approved the manuscript.

**Data availability statement.** Bright-field and phase image data used in this work may be made available to users upon reasonable request. A version of this article is available online as a preprint in arXiv:2210.04421 (Kumar et al., 2022).

**Supplementary material.** The supplementary material for this article can be found at https://doi.org/10.1017/qpb.2023.5.

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
