## [Reviewer Report]

Dear Editor,

We are very happy to submit the manuscript titled "QUANTIFICATION OF P OLLEN VIABILITY IN Lantana camara BY

DIGITAL HOLOGRAPHIC MICROSCOPY" as a research article to the Quantitiative Imaging Special Issue of your Journal. In this work, we present a novel combination of optics and palynology research, applied to plant Invasion Biology. We show how digital holographic microscopy (DHM) can be an efficient visual method for assessing the viability of pollen through quantitative phase information, which in turn can be correlated with chromatin content, thereby to viability. We successfully test this method on pollen grains of Lantana camara, a well-known alien invasive plant in most of the tropical world. Our results show that pollen viability can accurately be assessed without the usual staining procedure, and can be applied to a number of emerging areas in plant science.

We hope that QPB would find this MS acceptable to be published, we have suggested two reviewers who have expertise in both Optics and Botany, but you are welcome to edit/modify these if you wish.

Looking forward to a positive response from you,

Best Regards

Gitanjali Yadav

---

## [Reviewer Report]

Both the Reviewers raise significant issues concerning this manuscript but at the same time encourage the Authors to perform the revision. My recommendation is therefore major revision. The Authors need to take into account all of the Reviewers’ suggestions and address all the questions. Moreover, English needs to be improved.

---

## [Reviewer Report]

To

Dr. Olivier Hamant

Editor-in-Chief,

Quantitative Plant Biology

March 13, 2023

SUB: RESPONSE TO COMMENTS/DECISION ON MS: QPB-22-0026 AND RESUBMISSION

Dear Dr. Hamant,

We thank you for the opportunity to revise the manuscript QPB-22-0026, entitled “Quantification of Pollen Viability in Lantana camara by Digital Holographic Microscopy.” submitted for publication in QPB as a research article. We also thank the Associate Editor Dr. Dorota Kwiatkowska, for sharing the reviewer’s comments that were very insightful and constructive, enabling us to perform

a major revision and significantly improve our work and insights. In particular, we thank the reviewers for pointing out a new insight in the paper that we had not realized earlier. It had been suggested to re-analyse the data in the overlap region of the histogram and this investigation revealed to us the superiority of the DHM to be greater than not only phase contrast microscopy, but also manual estimation of viability! We have also improved the English in the paper and removed redundancy in text, and all

suggestions/modifications have been incorporated into the revised version, as follows:

1. Phase contrast microscopy was performed on the pollen samples and we were able to show that DHM provides quantification and performs better. 

2. The overlap region in the histogram was re-investigated to reveal that DHM may not just be 90% accurate (as we had initially thought), but much more, leading to new scope for future work and collaboration in the area.

3. We have incorporated 2 new Figures and Supplementary Materials data.

4. Text has been extensively modified along with new section headings to better reflect our rationale and conclusions.

5. Sections on Introduction, Methodology, Results and Conclusions have been modified in view of the reviewer’s comment and suggestions.

This revision is being submitted along with this cover letter, a detailed response to reviewers document, Supplementary Data and two versions of the revised MS; one marked with all edits and one unmarked version in PDF.

In light of the above, I sincerely hope that you will find the revised manuscript interesting and suitable for publication in Quantitative Plant Biology.

With best regards,

Gitanjali Yadav

Scientist, Computational Biology Laboratory,

NIPGR, Aruna Asaf Ali Marg

New Delhi, India - 110067

Email: gy@nipgr.ac.in

Tel: 91-11-26735103

---

## [Reviewer Report]

Both the Reviewers appreciate the improvement of the manuscript and point to only a few more changes that need to be performed. I thus recommend (very) minor revision. Please follow the Reviewers advice.

---

## [Reviewer Report]

To

Dr. Olivier Hamant

Editor-in-Chief, 

Quantitative Plant Biology

May 19, 2023

SUB: RESPONSE TO DECISION ON MS: QPB-22-0026 AND RESUBMISSION

Dear Dr. Hamant,

We thank you for acceptance of our manuscript QPB-22-0026, entitled “Quantification of Pollen Viability in Lantana camara by Digital Holographic Microscopy.” submitted for publication in QPB as a research article. 

We also thank the Associate Editor Dr. Dorota Kwiatkowska, for sharing the reviewer’s (minor) comments that have now been incorporated in the revised MS.

This final version is being submitted along with this cover letter, a detailed response to reviewers document, Supplementary Data and the final revised MS; one marked with all edits and one unmarked version in PDF. 

In light of the above, I thank you once again for finding the work interesting and suitable for publication in Quantitative Plant Biology.

With best regards,

Gitanjali Yadav

Scientist, Computational Biology Laboratory, 

NIPGR, Aruna Asaf Ali Marg

New Delhi, India - 110067

Email: gy@nipgr.ac.in

Tel: 91-11-26735103